# Refractory Salmonella Prosthetic Valve Endocarditis Complicated by Splenic Infarction and Aortic Pseudoaneurysm in a Patient with Double Prosthetic Valves: A Case Report

**DOI:** 10.3390/diagnostics12081982

**Published:** 2022-08-16

**Authors:** Moustafa S. Alhamadh, Rakan B. Alanazi, Thamer Saad Alhowaish, Abdulrahman Yousef Alhabeeb, Sultan T. Algarni, Osama Mohaamad Wadaan, Ihab Suliman, Mohammed Ghormalla Al-Ghamdi

**Affiliations:** 1College of Medicine, King Saud bin Abdulaziz University for Health Sciences (KSAU-HS), Ministry of the National Guard-Health Affairs, Riyadh 14611, Saudi Arabia; 2King Abdullah International Medical Research Center, Ministry of the National Guard-Health Affairs, Riyadh 11481, Saudi Arabia; 3Department of Adult Cardiology, King Abdulaziz Medical City Ministry of the National Guard-Health Affairs, Riyadh 11426, Saudi Arabia

**Keywords:** prosthetic valve endocarditis, *Salmonella* endocarditis, *Salmonellosis*, Commando procedure, splenic infarction, aortic pseudoaneurysm

## Abstract

Endocarditis is an extremely rare complication of *Salmonellosis* with an incidence of 0.2–0.4%. It is a destructive and invasive infection that follows a highly complicated course and carries a high mortality rate that exceeds 45%. Multiple predisposing factors for *Salmonella* endocarditis have been described in the literature, including human immunodeficiency virus infection, congenital heart diseases, and the presence of a prosthetic valve. Herein, we report a case of *Salmonella* prosthetic valve endocarditis complicated by splenic infarction and aortic pseudoaneurysm presenting as a month-long history of fluctuating fever, chills, and rigors, accompanied by occasional cough and shortness of breath in a 55-year-old female with aortic and mitral valves replacement and multiple comorbidities. She was diagnosed by multiple radiographic studies and successfully treated with the Commando procedure and a long course of IV antibiotics.

## 1. Introduction

*Salmonella* species are gram-negative motile facultatively anaerobic bacilli that commonly cause self-limited gastroenteritis [1]. It is transmitted via the ingestion of contaminated food and water or rattlesnake meat or is acquired via the fecal–oral route [2]. Although commonly benign, *Salmonellae* are responsible for 600 deaths annually, and up to 10% of the cases develop bacteremia, particularly among infants, young children, the elderly, and immunocompromised individuals, affecting vital systems such as the nervous and cardiovascular systems [3,4]. Endocarditis is an exceedingly rare complication of *Salmonella* infection with an incidence of 0.2–0.4% [5]. It is a fulminant, destructive, and invasive infection that follows a highly complicated clinical course with possible life-threatening complications, such as conduction disturbances, valvular abscesses, perforation and/or rupture, and carries a high mortality rate that exceeds 45% [5,6,7,8]. Several *Salmonella* serotypes have been implicated in infective endocarditis, the most common of which are *S. choleraesuis*, *S. typhimurium*, *S. Thompson*, and *S. typhi* [2]. Mitral valve *Salmonella* infective endocarditis comprises 33.3% of *Salmonella* infective endocarditis cases, whereas *Salmonella* mural (nonvalvular) infective endocarditis comprises 26.4%, with a higher mortality rate of 52% [2]. Multiple predisposing factors for *Salmonella* endocarditis have been described in the literature, including human immunodeficiency virus (HIV) infection, congenital heart diseases, and rheumatic heart disease (RHD) and subsequent prosthetic valve replacement [5,9]. 

## 2. Case Presentation

A 56-year-old Saudi female, with a known case of type 2 diabetes mellitus, hypertension, chronic anemia, atrial fibrillation, non-ischemic cardiomyopathy, and RHD status post mitral and aortic valves replacement, presented to our emergency department complaining of a month-long history of fluctuating fever, chills, and rigors, accompanied by occasional cough and mild shortness of breath. At first, she had a sore throat, cough, and sneezing, for which she visited a private clinic and was diagnosed as a case of tonsillitis and prescribed a short course of antibiotics without significant improvement. She denied a history of sick contact, chest pain, palpitations, syncope, abdominal or joint pains, urinary or bowel changes, or nausea and vomiting. Upon admission, she was on digoxin and furosemide for heart failure, warfarin for atrial fibrillation, bisoprolol and candesartan for hypertension, insulin glargine for diabetes mellitus, rosuvastatin for dyslipidemia, and esomeprazole for occasional reflux. Her past surgical history was remarkable for a MAZE procedure a year ago for atrial fibrillation and mechanical aortic (size 21 mm) and mitral (size 31 mm) valves replacement for RHD 2 years ago. Her family history was noncontributory. On examination, she was ill-appearing, normotensive (115/67 mmHg), and tachycardiac (140 bpm) with a high-grade fever (39.5 °C). The clinical examination of the respiratory system revealed the presence of right basal crepitations. Cardiovascular examination was unremarkable, with normal S1 and S2 with appreciable metallic clicks, and no added sounds or murmurs, carotid or femoral bruits, or thrills or heaves. Her abdomen was nontender and soft, with normal bowel sounds, and without evidence of organomegaly. A trace right lower limb edema was observed during examination. She was admitted under cardiology for diagnosis and management as appropriate.

Laboratory investigations were notable for normocytic normochromic non-hemolytic anemia (Hgb: 115 g/L, MCV: 85.8 fL, MCHC: 330 g/L, Haptoglobin: 1.87 g/L), thrombocytopenia (Platelets: 96 × 10^9/L), abnormal liver function (ALT: 57 U/L, AST: 113 U/L, Alk Phos: 188 U/L, Total Bili: 55.9 μmol/L, GGT: 292 U/L), kidney injury (Creatinine: 264 μmol/L, BUN: 23 mg/dL, eGFR: 17 mL/min/1.73 m^2^), elevated uric acid (655 μmol/L), lactic acid (2.42 mmol/L), inflammatory markers (ESR: 85 mm/h, CRP: 336 mg/L, PCT: 23.7 ng/mL), cardiac biomarkers (BNP: 256 pmol/L, Troponin I: 75 pg/mL), thrombotic markers (D-dimer: 12.32 mg/L, Fibrinogen: 6.23 g/L), hypotonic hyponatremia (170 mOsm/kg, 119 mmol/L), hypochloridemia (88 mmol/L), and hypoalbuminemia (30 g/L). Her coagulation profile was remarkable for elevated PT (59.1 s), PTT (45 s), and INR (6.62) (Table 1). Urinalysis was unremarkable, and blood cultures came positive for *Salmonella* (group C and D) three times during the patient’s admission, despite appropriate IV antibiotics. 

Chest radiograph was notable for bilateral interstitial infiltrates representing pulmonary edema (Figure 1), and admission ECG showed atrial fibrillation with a rapid ventricular response (Figure 2). A cardiac positron emission tomography/computed tomography (PET/CT scan) revealed increased Fluorodeoxyglucose (FDG) uptake around the prosthetic aortic valve and the aortic root, indicating an ongoing inflammatory or infectious process (Figure 3 and Figure 4). Cardiac CT findings were periaortic collection, extending to the anterior mediastinum along the retrosternal area, and associated with enhancing wall, mediastinal stranding, and multiple enlarged lymph nodes. Furthermore, there was a pseudoaneurysm in the ascending aorta (Figure 5). Transesophageal echocardiogram was performed and showed a posterior aortic root abscess cavity extending to the aortomitral curtain, bulging in the left atrium, and communicating with the left ventricular outflow tract. Moreover, there was moderate concentric hypertrophy, global hypokinesis of the left ventricle with an ejection fraction of 45%, and mild tricuspid regurgitation (Appendix A). After a week of admission, she developed epigastric pain, for which she underwent abdominal and pelvis CT. It demonstrated a large hypodense area in the lower pole of the spleen with surrounding fat stranding and another upper pole linear hypodensity, representing infected splenic infarction (Figure 6). A diagnosis of complicated *Salmonella* aortic prosthetic valve endocarditis was made, and the patient was counselled about the risk of surgical intervention, and she agreed. 

The patient underwent a Commando procedure in which a redo sternotomy with peripheral cardiopulmonary bypass was established, infected tissues were resected, mitral valve was replaced with a 27 mm mechanical valve, aortomitral curtain and left atrial roof were reconstructed, aortic root and ascending aorta were replaced, and modified Valsalva conduit and 21 mm tissue aortic valve were implanted. Following that, she was admitted to the cardiac intensive care unit (ICU) and commenced on IV Ceftriaxone and Gentamicin. A few days after ICU admission, she became severely hypotensive and developed acute kidney injury requiring continuous renal replacement therapy in the form of hemodialysis (HD). Two weeks postoperatively, a follow-up transthoracic echocardiography demonstrated well-seated aortic and mitral valves, mild aortic regurgitation, and localized posterior pericardial effusion. 

After 18 days of ICU admission, she was shifted to the floor and commenced on conventional HD via perma-cath thrice weekly. She was discharged after 4 months of admission, asymptomatic and hemodynamically stable with a negative septic workup. On follow-up 2 months later, she stopped HD, and the perma-cath was removed. Overall, she was healthy, with good urine output and stable kidney function (Creatinine: 149 μmol/L, BUN: 15.9 mg/dL) (Appendix A). 

## 3. Discussion

*Salmonella* species are Gram-negative bacilli that commonly cause self-limited gastroenteritis but may infect the meninges, bones, myocardium, and other vital organs [1,3,4]. Endocarditis is an exceedingly rare complication of *Salmonella* infection with an incidence of 0.2–0.4% and a high mortality rate even with appropriate antibiotics therapy [5]. It is a fulminant, destructive, and highly invasive disease that is possibly complicated by conduction defects, valvular abscesses, perforation, and/or rupture [6,7,8]. It usually occurs in patients with HIV and preexisting heart diseases, including congenital heart diseases and RHD [5,9]. In this article, we report a case of Salmonella prosthetic valve endocarditis complicated by splenic infarction, prolonged hospital stay, and kidney injury in a 56-year-old female with multiple cardiovascular comorbidities.

We believe that this case is interesting for the following reasons. (1) This case highlights the importance of maintaining a high index of clinical suspicion for *Salmonella* endocarditis in patients with prosthetic valves who present with bacteremia, especially in the absence of extracardiac sources of infection such as IV catheters and pneumonia. Early recognition and surgical intervention are essential to decrease *Salmonella* endocarditis morbidity, such as conduction abnormalities, septic embolism, valvular abscesses, perforation or rupture, and mortality [10]. (2) To the best of our knowledge, this might be the second case of *Salmonella* prosthetic valve endocarditis in the Kingdom of Saudi Arabia, but the first case of aortic valve *Salmonella* endocarditis, as the previously reported case was Salmonella mitral valve endocarditis [11]. (3) This case is unique because the patient had double prosthetic valves (mitral and aortic), and her case was uniquely complicated by splenic infarction due to septic emboli, aortic pseudoaneurysm, and aortic root abscess cavity extending to the aortomitral curtain, bulging in the left atrium, and communicating with the left ventricular outflow tract. This might be the first case of aortic pseudoaneurysm as a complication of *Salmonella* prosthetic valve endocarditis. Although cerebral embolism is a known complication of *Salmonella* endocarditis [12,13,14,15], splenic infarction due to emboli in the setting of *Salmonella* endocarditis has only been reported once [16]. (4) The patient’s age is atypical for salmonella endocarditis, as most of the reported cases were younger than 50 years [4]. (5) *Salmonella* prosthetic valve endocarditis is extremely rare, and to the best of our knowledge, only around 20 cases have been reported in the literature.

The Modified Duke criteria consist of clinical, laboratory, and imaging findings that are used in the diagnosis of endocarditis. The diagnosis of endocarditis requires two major criteria, one major and three minor criteria, or five minor criteria [17]. In this case, three blood cultures came positive for Salmonella during the patient’s admission and PET/CT with FDG and cardiac CT findings were highly suggestive of endocarditis. Based on the European Society of Cardiology, findings suggestive of endocarditis on ECG-gated cardiac CT angiogram or PET/CT with FDG are considered major criteria [17]. Additionally, the reported case meets some of the minor criteria, such as having a predisposing heart condition and persistent fever of ≥38.0 °C (≥100.4 °F).

*Salmonella* is an intestinal pathogen, but it can invade the epithelium, flourish intracellularly, and, through the lymphatic system, spread systemically [5]. Nontyphoidal *Salmonella* is of particular interest, as it can invade the arterial intima and therefore infect the endothelium in the presence of atherosclerosis through a poorly understood mechanism [18]. In our patient, multiple blood cultures came positive for nontyphoidal *Salmonella* (group C and D), which could explain her aortic involvement. Based on the few reported cases, it seems that patients with HIV or preexisting cardiac disease, especially those aged ≥ 50 years, might be more vulnerable to *salmonella* endocarditis [5,9], but it has been reported in normal heart valves [19,20]. Our patient had a history of RHD previously treated by double prosthetic valve replacement, atrial fibrillation, and cardiomyopathy.

Old data showed poor efficacy for antibiotics in the treatment of *Salmonella* endocarditis [10], but more recent data showed that third generation cephalosporins, such as Ceftriaxone, and fluoroquinolones are often effective [5]. Besides surgery, our patient was treated with IV Ceftriaxone and Gentamicin. Although highly complicated and risky, surgical intervention has been shown to greatly improve survival and should be considered in patients with extensive disease [21].

## 4. Conclusions

*Salmonella* prosthetic valve endocarditis represents a small fraction but a life-threatening form of *Salmonella* endocarditis. Since unified diagnostic criteria are lacking, *Salmonella* endocarditis represents a diagnostic challenge, and clinicians should maintain a low threshold of its diagnosis, especially in patients with preexisting heart disease, as early medical and surgical interventions can prevent or mitigate its associated morbidity and mortality. 

## Figures and Tables

**Figure 1 diagnostics-12-01982-f001:**
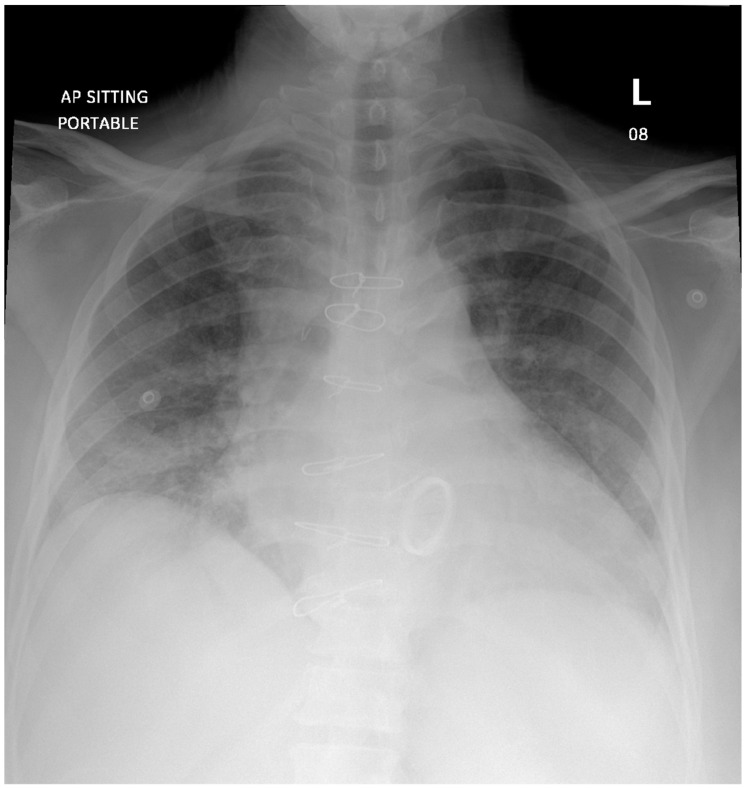
Chest X-ray showing bilateral airspace opacities and interstitial infiltrates representing pulmonary edema.

**Figure 2 diagnostics-12-01982-f002:**
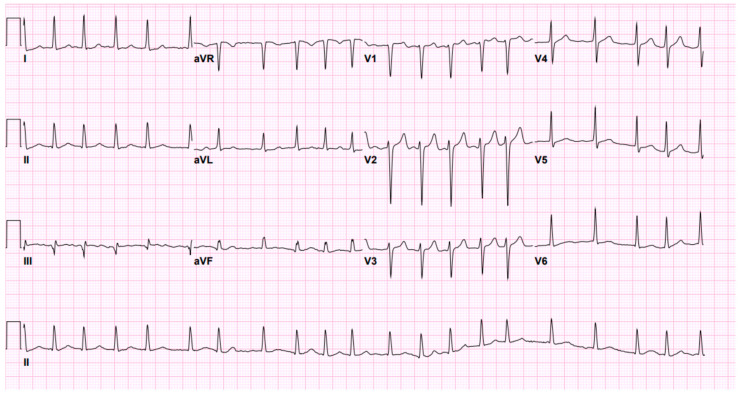
ECG showing atrial fibrillation with a rapid ventricular response.

**Figure 3 diagnostics-12-01982-f003:**
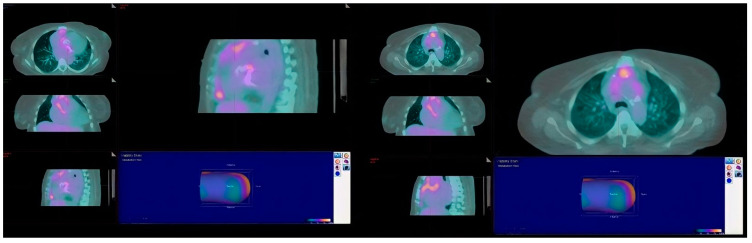
Cardiac PET scan showing increased FDG uptake around the prosthetic aortic valve and the aortic root, indicating an ongoing inflammatory or infectious process.

**Figure 4 diagnostics-12-01982-f004:**
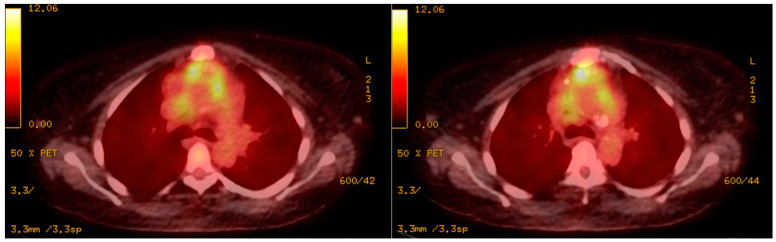
Cardiac CT scan showing increased FDG uptake around the prosthetic aortic valve and the aortic root, indicating an ongoing inflammatory or infectious process.

**Figure 5 diagnostics-12-01982-f005:**
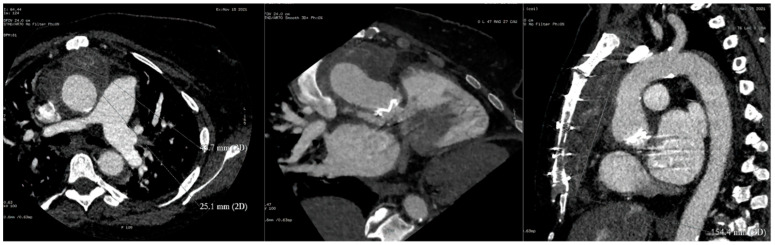
Cardiac CT showing periaortic collection and pseudoaneurysm in the ascending aorta.

**Figure 6 diagnostics-12-01982-f006:**
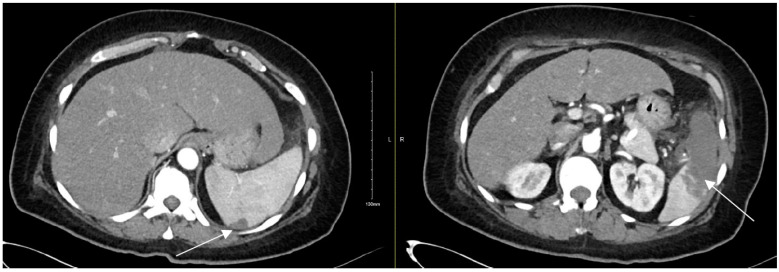
Abdominal and pelvis CT showing a large hypodense area in the lower pole of the spleen with and another upper pole linear hypodensity, representing infected splenic infarction. The white arrows point at hypodense areas indicating splenic infarction.

**Table 1 diagnostics-12-01982-t001:** Important Laboratory Values at Hospital Admission, ICU Admission/One-Day Preoperatively, Discharge, and Follow-Up.

Test Name	Reference Range with Unit	At Initial Admission	At ICU Admission	At Discharge	At Follow-Up
Hgb	120~160 gm/L	115	88	100	98
MCV	76~96 fL	82.8	93.6	93.9	93.1
Retic Percent	0.5~1.5%	5.6	5.08	N/A	3.37
RBC	4~5.4 × 10^12^/L	4.21	2.84	3.16	3.29
WBC	4~11 × 10^9^/L	10.2	8.25	7.32	6.88
Neutrophils	2~7.5 × 10^9^/L	9.32	6.28	5.01	5.45
Lymphocytes	1~4.4 × 10^9^/L	0.3	1.01	1.2	1.68
Platelets	150~400 × 10^9^/L	96	219	288	212
PT	Second	55.9	12.2	36.1	26.3
PTT	Second	41.9	41.6	36.5	31
INR	.	6.21	1.14	3.6	2.54
AST	5~34 U/L	113	28	52	34
ALT	5~55 U/L	57	14	28	24
Bili T	~20.5 μmol/L	55.9	39.5	38.8	28.1
Bili D	~8.6 μmol/L	40	30	23.3	16
GGT	9~36 U/L	290	286	576	627
Alk Phos	40~150 U/L	188	281	193	199
Total Protein	64~83 g/L	65	66	82	77
BUN	3.5~7.2 mmol/L	23	13.6	9.9	15.9
Creatinine	50~98 μmol/L	264	190	213	149
eGFR	~60 mL/min/1.73 m^2^	17	25	22	25
Albumin	35~50 g/L	30	29	47	48
Uric Acid	150~370 μmol/L	655	363	257	701
Sodium	136~145 mmol/L	119	109	129	136
Chloride	98~107 mmol/L	88	80	94	102
Phosphorus	0.74~1.52 mmol/L	1.18	1.27	0.87	1.62
Ca	2.1~2.55 mmol/L	1.86	1.87	2.41	2.33
Adj Ca	2.1~2.55 mmol/L	2.06	2.09	2.27	2.17
Potassium	3.5~5.1 mmol/L	4.6	4.7	5.1	5.1
Magnesium	0.66~1.07 mmol/L	0.93	0.8	0.8	0.8
Lactic Acid	0.5~2.2	2.96	2.58	N/A	N/A
CRP	~8 mg/L	336	89	N/A	N/A
PCT	~0.05 ng/mL	23.7	19.5	N/A	N/A
ESR	0~30 mm/h	85	59	N/A	N/A
D-Dimer	0~0.5	12.32	4.9	N/A	N/A
BNP	~28.9 pmol/L	256	420	N/A	295
Troponin I	~15.6	75	2196	N/A	N/A
Creatine Kinase	29~168 U/L	151	455	N/A	N/A

Abbreviations N/A: Not available, Hgb: Hemoglobin, MCV: Mean Corpuscular Volume, RBC: Red Blood Cells, WBC: White Blood Cells, PT: Prothrombin Time, PTT: Partial Thromboplastin Time, INR: International Normalized Ratio, AST: Aspartate transaminase, ALT: Alanine aminotransferase, Bili T: Total Bilirubin, Bili D: Direct Bilirubin, GGT: Gamma-glutamyl Transferase, Alk Phos: Alkaline Phosphatase, BUN: Blood Urea Nitrogen, GFR: Glomerular Filtration Rate, Ca: Calcium, CRP: C-Reactive Protein, PCT: Procalcitonin, ESR: Erythrocyte Sedimentation Rate, BNP: Brain Natriuretic Peptide.

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
