# Peer review of "Refractory Salmonella Prosthetic Valve Endocarditis Complicated by Splenic Infarction and Aortic Pseudoaneurysm in a Patient with Double Prosthetic Valves: A Case Report"

_diagnostics, 2022, doi:10.3390/diagnostics12081982_

Round 1

Reviewer 1 Report

This manuscript focuses on prosthetic valve endocarditis of very rare etiology. Although the complications were multiple and severe, the case management was successful. Sharing the author’s experience is important for clinical practice.  I appreciated the imagistic also.

However, some improvements are necessary.

Introduction I suggest „conduction disturbances” instead of “conduction defects”

The Introduction is too summary. I recommend adding information about Salmonella serotypes known to cause endocarditis, valves usually involved, the clinical aspect encountered in Salmonella endocarditis and the treatment adopted (the scientific knowledge currently available).

Case:

“and RHD status post mitral and aortic valve replacement”. Please verify this statement. Valve replacement is necessary after RHD.

Her past surgical history was remarkable for MAZE procedure… I recommend adding information about the valve prosthesis: type of prosthesis, age of the prosthesis.

The patient’s chest was remarkable for right basal crepitations.  I suggest “the clinical examination of the respiratory system revealed the presence of right basal crepitations”

transaminitis (ALT: 57 U/L, AST: 113 U/L, Alk Phos: 188 74 U/L, Total Bili: 55.9 umol/L, GGT: 292 U/L). Not all parameters represent transaminitis. I suggest abnormal liver function.

Discussion: In the presented case the digestive symptoms were absent. I invite the authors to comment on this aspect. Was the stool exam performed in this case? Was Salmonella portage considered at any time in the case management?

Of 19 references, only four are from the last 10 years. The newest is from 2016. I recommend the authors update the manuscript, as advances in this field are important for practice.

Thank you!

Author Response

On behalf of the authors, I would like to thank you for your honest review and constructive feedback. Your suggestions will definitely improve the paper’s quality.

Response to comment 1:

“Conduction defects” was changed into “conduction disturbance”.

Response to comment 2:

I totally agree with you. The introduction is too short. We added more information regarding Salmonella serotypes that are known to cause endocarditis and the treatment outcomes.

Response to comment 3:

We added more information regarding the valves replacement in the past medical/surgical section. We wrote, “Her past surgical history was remarkable for MAZE procedure a year ago for atrial fibrillation and mechanical aortic (size 21 mm) and mitral (size 31 mm) valves re-placement for RHD 2 years ago.”

Response to comment 4:

As mentioned in the previous response, we added the size of the mechanical valves and the time of the surgery, which was 2-3 years ago. Unfortunately, I could not find the type (name) of the valves.

Response to comment 5:

As suggested, “chest was remarkable for right basal crepitations” was changed into “the clinical examination of the respiratory system revealed the presence of right basal crepitations”.

Response to comment 6:

You are totally correct. We changed transaminitis into abnormal liver function.

Response to comment 7:

Yes, there were no GI/digestive symptoms at the time of the presentation. I called the patient to double check and she confirmed the absence of GI/digestive symptoms. To my knowledge, a stool exam was not performed.

Response to comment 8:

We extensively reviewed the literature, but we could not find recent cases regarding the topic. As discussed in the discussion section, this is an extremely rare pathology, and only 20 cases were published before. We did another search when we saw your comment, and we could not find recent papers. There are more recent papers on salmonella endocarditis but not on prosthetic valves.

Thank you again for your comments and have a lovely day,

Moustafa S. Alhamadh

Reviewer 2 Report

This is a very interesting manuscript describing a patient with a prosthetic valve Infective Endocarditis complicated by pseudoaneurysm and infarction due to Salmonella. I have some comments that can be found below:

1.       Line 22 and 48: human immunodeficiency virus infection

2.       Line 23: presence of a prosthetic valve

3.       Line 32: Please remove the word, figure etc counting from the manuscript

4.       Line 53: Please provide the information when were the prosthetic valves placed, if it is available

5.       Line 60: no need for writing the medications with capital letters

6.       Line 64: Not sure if this pressure justifies the word ‘hypotensive’

7.       Line 72 and on: It would be better to explain all abbreviations when first used. For example: procalcitonin (PCT), erythrocyte sedimentation rate (ESR). This applies for AST, ALT, CT, INR, BUN etc)

8.       Table 1: Please define all abbreviations used in a footnote

9.        Line 126: You state here that the patient has chronic kidney disease. That was not mentioned at the beginning of the case presentation. Please correct that, and if possible provide the baseline creatinine value

10.   The basic correction I would like to see is the antibiogram of the microorganism. I would like to see the antimicrobial susceptibility pattern provided either in the main manuscript (preferably) or in the supplementary material

11.   Line 83: What antibiotics were used at that point?

12.   Line 124: what was the dose of the antimicrobials?

13.   Was the dose of gentamicin adjusted based on blood levels post administration?

14.   Line 190: Not sure if an abbreviation section is needed. You could introduce each abbreviation in the abstract, the tables and in the main text when first used

15.   We completely agree this is an informative case of endocarditis. However, please provide a statement mentioning the Dukes criteria that typically confirm the diagnosis. (eg 2 major and 3 minor)

Author Response

On behalf of the authors, I would like to thank you for your honest review and constructive feedback. Your suggestions will definitely improve the paper’s quality.

Response to comment 1:

Human immunodeficiency virus was changed into human immunodeficiency virus infection.

Response to comment 2:

We added the “presence of a prosthetic valve” as suggested.

Response to comment 3:

The word, images, and video count were removed as requested.

Response to comment 4:

We added more information regarding the valves replacement in the past medical/surgical section. We wrote, “Her past surgical history was remarkable for MAZE procedure a year ago for atrial fibrillation and mechanical aortic (size 21 mm) and mitral (size 31 mm) valves re-placement for RHD 2 years ago.” Also, more information was added in the introduction.

Response to comment 5:

We changed the capital letters into small letters in the medication list as suggested.

Response to comment 6:

Yes, I totally agree with you. This was a mistake. It was changed into normotensive.

Response to comment 7:

I tried to write the full word before any abbreviations, but it looked confusing when I wrote the full name of each laboratory tests/parameters (AST, ALT, CT, INR, BUN …). I explained all the abbreviations below table 1.

Response to comment 8:

We defined all the abbreviations below the table. 

Response to comment 9:

I added a figure showing the creatinine level from 2021 to the most recent one. Please give me your opinion on whether to keep the figure or delete it.

Response to comment 10:

Unfortunately, I could not get the antibiogram and antimicrobial susceptibility pattern. As per the report, it was susceptible to gentamicin, ampicillin, and TMP/SMX if I remembered correctly.

Response to comment 11:

The documentation is a bit confusing, but she was on multiple antibiotics including Vancomycin and Ceftriaxone (7 days if the documentation was correct). Then Vancomycin was discontinued, and she was put on Gentamicin. Before starting vancomycin, she was on Tazocin.

Response to comment 12:

Ceftriaxone (2000 mg – q24h over 3-5 min) and Gentamicin (Initially it was 80 mg – q12h over 30-60 min, then the clinical pharmacist change it into 120 mg – q24h over 30-60 min)

Response to comment 13:

I could not find information regarding this. I apologize.

Response to comment 14:

We removed the abbreviations section as suggested.

Response to comment 15:

Thank you for bringing this up. I totally agree with you, this is an important point that should be discussed in the discussion. I added some information regarding Duke criteria and how this case meets it.

Thank you again and we really appreciate your review. Have a lovely day,

Moustafa S. Alhamadh

Round 2

Reviewer 2 Report

The manuscript has been improved during the revision process. Figure 7 can be deleted or added in the supplementary material. However, one can see in the diagram, the patient's creatinine before admission was normal, thus, the statement that she had chronic kidney disease should be probably deleted.

Author Response

Thank you for your quick response, we appreciate it.

As requested, we deleted the statement that says she had chronic kidney disease.

Figure 7 will be submitted as supplementary material.

Thank you again and have a lovely day,

Moustafa